# Comparative Effectiveness of Multi-Component, Exercise-Based Interventions for Preventing Soccer-Related Musculoskeletal Injuries: A Systematic Review and Meta-Analysis

**DOI:** 10.3390/healthcare13070765

**Published:** 2025-03-29

**Authors:** Yucheng Wang, Zhanyi Zhou, Zixiang Gao, Yaodong Gu

**Affiliations:** 1Faculty of Sports Science, Ningbo University, Ningbo 315211, China; 2411040024@nbu.edu.cn (Y.W.); zhouzhanyi630@hotmail.com (Z.Z.); 2Human Performance Laboratory, Faculty of Kinesiology, University of Calgary, Calgary, AB T2N 1N4, Canada

**Keywords:** soccer, injury prevention, musculoskeletal injuries, systematic review, meta-analysis

## Abstract

**Background:** Soccer is a high-intensity sport characterized by a considerable incidence of injuries, particularly among professional male players, with injury rates ranging from 5.9 to 9.6 per 1000 player-hours. Lower limb injuries, including those affecting the knee, ankle, hip/groin, and hamstring muscles, are particularly prevalent. Additionally, a history of prior injuries may exacerbate the risk of recurrence. In response to these concerns, various injury prevention programs have been developed and implemented, targeting different genders and age groups. **Methods:** This systematic review and meta-analysis, conducted with the PRISMA guidelines, critically evaluated randomized RCTs across diverse genders and age groups to assess the efficacy of multi-component exercise-based injury prevention programs in reducing musculoskeletal injuries among soccer players. Comprehensive searches were conducted in ClinicalTrials.gov, CENTRAL, EMBASE, PubMed, Scopus, and Web of Science, with no language restrictions applied. **Results:** A total of 15 RCTs met the predefined inclusion criteria. The intervention programs were found to be significantly effective in reducing lower limb injuries, with a pooled RR of 0.73 (95% CI: 0.63 to 0.84, *p* = 0.035). Subgroup analyses further revealed a significant reduction in the incidence of hamstring, knee, and ankle injuries following the implementation of these programs. **Conclusions:** Multi-component exercise-based injury prevention programs demonstrate considerable efficacy in reducing musculoskeletal injuries in soccer players, spanning various age groups and genders. These findings underscore the potential of such programs in professional soccer injury management and highlight their importance in the development of comprehensive injury prevention strategies.

## 1. Introduction

In professional men’s soccer, injuries are frequent, occurring at rates of 5.9 to 9.6 per 1000 player-hours [1,2]. The sport’s fast-paced actions, like sprints, quick stops, and sudden turns, heighten the risk of muscle and joint injuries. Prolonged or repeated injuries can shorten careers, with studies showing that injuries account for up to 30% of early retirements [3]. As such, recent studies have aimed to identify specific muscles and joints that may be of high risk in this population. Gibbs [4] found that quadriceps muscle strain is relatively common during sprinting or powerful kicking movements, primarily occurring during the eccentric contraction phase. In this phase, the muscle undergoes passive lengthening while simultaneously generating tension to control the movement, requiring it to withstand a high mechanical load, increasing the risk of muscle fiber injury. Saw et al. [5] noted hamstring injuries often occur during deceleration as the muscle stretches to slow movement. Elliott et al. [6] highlighted calf muscles, especially the gastrocnemius and soleus, as frequently injured during kicking and pivoting. Gaulrapp et al. [7] reported ankle sprains as a major issue during high-impact actions like landing or sharp turns, accounting for a significant share of soccer injuries. Other studies have indicated that the knee, ankle, and other lower limb joints have a high incidence of injuries [8,9,10].

Although the precise mechanisms underlying these injury patterns remain incompletely understood, evidence indicates that a history of previous injuries substantially increases the probability of future injury occurrence. Consistent with epidemiological patterns reported in soccer athletes, current findings corroborate the predominance of ligamentous involvement specific to knee (i.e., 17–23%) [11] and ankle injuries (i.e., 20–28%) [12]. The observed hamstring injury prevalence of 12–15% (i.e., annual incidence) aligns with established risk ratios ranging from a 2.5 to 3.8 fold-increase, likely attributable to the cumulative effect of high-velocity eccentric loading during sprint cycles [13]. Meanwhile, previous studies have indicated that the injury rate among amateur athletes is generally lower than that of professional players [14,15,16].

In addition to non-contact injuries, contact-related injuries also play a significant role in the injury profile of soccer players. These injuries occur during physical contact with other players, such as tackles, and contribute to the overall injury burden in the sport. The rate of contact-related injuries varies, but studies have shown that they are a major cause of injuries in both professional and amateur soccer [17]. Improving physical capacity, including strength, flexibility, and stability, can reduce the likelihood of both contact and non-contact injuries. In contact sports like soccer, stronger muscles are better able to absorb impacts, which helps prevent injuries caused by sudden physical confrontations [18,19].

However, strengthening muscles alone is not sufficient to fully mitigate injury risks as multiple factors contribute to both contact and non-contact injuries. Addressing these risks requires a more comprehensive approach that enhances various physical attributes essential for injury prevention. Multi-component exercise-based injury prevention programs outperform traditional single-focus training by integrating diverse exercises that improve overall health, athletic performance, muscle balance, flexibility, and stability, effectively reducing injury risks across all genders and age groups in soccer [20,21,22,23,24,25]. Soligard et al. [17] introduced multi-component exercise-based injury prevention programs combining different exercise types to reduce injury risks. This approach improves physical capacities like strength, flexibility, balance, and proprioception, which enhances overall function and reduces injuries. By addressing issues such as muscle imbalances, poor neuromuscular control, and joint instability, these programs target key injury causes. Examples include: (1) Strength training, which builds muscle resilience to prevent strains [18]; (2) Balance exercises, which enhance joint stability and proprioception, reducing sprains [19]; and (3) Plyometric drills, which improve force absorption and distribution, lowering overuse injury risks.

However, isolated training often overlooks the synergistic effects among various physical qualities. During physical activity, qualities such as strength, balance, and proprioception interact with each other and are all indispensable. For instance, if proprioception of the knee joint is excellent but the hip muscles are weak, knee stability may not be effectively supported. Conversely, individuals with strong hip muscles but without good proprioceptive input cannot fully utilize the incoming sensory information to appropriately regulate and output movement. Therefore, isolated training fails to fully activate the synergistic effects of the body’s various components, whereas multi-component exercise-based injury prevention programs can enhance multiple physical qualities through the integration of various exercises, promoting better coordination and development of all physical abilities. By combining these exercises, athletes’ musculoskeletal systems are strengthened, allowing them to better adapt to the demands of physical activity [26]. By integrating strength, balance, and plyometric exercises, these programs offer a multi-layered preventive approach that supports passive structures while enhancing active neuromuscular control, ultimately increasing overall resilience against injuries.

Despite the development of injury prevention programs, there is a lack of meta-analyses covering randomized controlled trials (RCTs) across genders and age groups. Additionally, research on programs targeting lower limb function and musculoskeletal injury risk in soccer players is limited. As such, the lack of literature demonstrates the need for more studies to guide evidence-based practices, reduce injury rates, and enhance player health and longevity. The aim of this systematic review and meta-analysis was to investigate the effects of injury prevention strategies and provide unique insights into the role of tailored interventions, offering a deeper understanding of how specific strategies can be optimized for various populations (i.e., age groups and gender groups).

## 2. Materials and Methods

### 2.1. Search Strategy and Selection Criteria

In conducting this systematic review and meta-analysis, we adhered to the PRISMA guidelines [27] and it has been registered in PROSPERO (CRD42024625910). A comprehensive search was performed across multiple databases, including ClinicalTrials.gov, CENTRAL, EMBASE, PubMed, Scopus, and Web of Science. The search terms included “soccer”, or “football”, combined with terms related to injury prevention, such as “injury prevention”, “warm-up program”, or “neuromuscular program”. The search was conducted from the inception of the databases up to 29 October 2024, with no restrictions on language. This study did not require ethical approval because this study was a systematic review and meta-analysis.

### 2.2. Study Screening and Data Extraction

The following studies were included if they met the inclusion criteria: (1) Studies that focused on multi-component injury prevention programs that included at least two distinct types of exercise, such as strength training, balance exercises, flexibility exercises, and plyometric drills; (2) Programs that aimed to improve physical capacities related to injury prevention, including muscle strength, joint stability, flexibility, and proprioception; (3) The intervention specifically targeted injury prevention rather than rehabilitation; (4) The study involved soccer players (both professional and amateur athletes) from various age groups and genders; (5) The study reported one of the following outcomes: the number of any lower limb injuries, including knee, ankle, and hamstring injuries.

The exclusion criteria were as follows: (1) Studies that involved participants who were not soccer players or those from other sports or non-athlete populations; (2) Research that focused on rehabilitation rather than injury prevention; (3) Studies that did not integrate at least two exercise modalities or included only one type of exercise component; (4) Non-randomized RCTs, case reports, reviews, and animal studies.

To ensure the completeness of the study and the reliability of the results, two reviewers independently extracted the following items from the included studies: (1) First author, publication year, age, the proportion of sample size for each group; (2) Interventions and control groups; (3) Outcomes, including lower extremity injuries, knee injuries, ankle injuries and hamstring injuries.

Any disagreements were resolved through consensus, and if necessary, a third reviewer made the final decision. If essential data were missing or unavailable in the published sources, we contacted the authors twice to request the required information.

### 2.3. Risk of Bias Assessment and Quality Evaluation

The risk of bias in the included studies was assessed by two independent researchers, Reviewer 1 and Reviewer 2 (Y.W. and Z.Z.), who were blinded to the study details, using the Cochrane Risk of Bias Tool (ROB 2.0) [28,29]. The assessment was conducted across the following five dimensions: (1) Bias from the randomization process; (2) Bias due to deviations from intended interventions; (3) Bias from missing outcome data; (4) Bias in outcome measurement; and (5) Bias in the selection of reported results. This process involved comparing the primary outcomes of the included studies with those of the current systematic review and network meta-analysis. Subsequently, the results were verified by a third reviewer (Z.G.) to ensure consistency and accuracy. The risk of bias for each included trial was assessed following the guidelines in the Cochrane Handbook for Systematic Reviews of Interventions [30]. Each domain was evaluated using the standardized 5-step process of the Cochrane Risk of Bias Tool (ROB 2.0): (1) Articulating bias sources; (2) Developing domain-specific criteria; (3) Implementing risk judgment (“Low”, “Some concerns”, or “High”) based on reporting completeness; (4) Substantiating judgments with evidence anchors; and (5) Formulating domain-level conclusions. The bias domains evaluated included sequence generation, allocation concealment, blinding of outcome assessment, completeness of outcome data, potential selective reporting, and other possible sources of bias. Each domain was categorized as either “low risk” or “high risk” or “some concerns” based on its potential impact on the study’s results. When a domain’s impact was considered likely, we examined whether the reported details were unclear or insufficient [31]. The two reviewers independently performed the risk of bias assessment, and any discrepancies were resolved through consultation with the third reviewer.

### 2.4. Data Extraction and Administration

For each eligible study, three reviewers independently extracted the data using a standardized form, following the methods of Thorborg et al. [32]. These data included fundamental study details, design characteristics, participant demographics, intervention components, and measured outcomes. The reviewers then compared their extracted data to ensure consistency. Any discrepancies were discussed, and if necessary, a fourth reviewer (Y.G.) was involved; final decisions were determined by a majority vote. Primary outcome data for each study were subsequently organized and recorded in Microsoft Excel (Microsoft 365, Version 2309; Microsoft Corporation, Redmond, WA, USA).

### 2.5. Data Analysis

The primary outcome measure was the incidence of lower limb injuries, while knee, ankle, and hamstring injuries were analyzed as secondary outcomes. These injuries were selected due to their high prevalence in soccer, where they are frequently cited in the literature as major contributors to musculoskeletal injuries [33,34,35,36,37]. Previous studies have highlighted knee and ankle injuries—particularly sprains and ligament strains—as common occurrences. These injuries often result from the sport’s physical demands, including rapid direction changes, jumping, and landing [4,5,7]. Hamstring injuries are also prevalent, especially during high-speed running or sudden deceleration, which places excessive strain on that muscle group [6,10]. Given their high incidence and significant impact on players’ performance and career longevity, these injuries were identified as key outcomes for this analysis.

Because all included studies were RCTs, the outcomes, including the number of extremity injuries, hamstring injuries, knee injuries, and ankle injuries were expressed as risk ratios (RR) with 95% confidence intervals (CIs). All statistical analyses were performed using RevMan software (version 5.4; Cochrane Collaboration, 2020), STATA 16, and R (version 4.3.0; R Core Team, 2023) [38]. Egger’s test was used to assess potential publication bias, with statistical significance set at *p* < 0.05 [39], and funnel plot asymmetry was visually inspected to further evaluate potential bias in the included studies [40].

### 2.6. Research Data Time Limitations

In this systematic review and meta-analysis, the included studies were assessed for their relevance and quality. The oldest study included in this review dates back to 2008. It is important to note that no specific time restrictions were applied to the studies included in this review. However, a broad range of studies was considered to ensure comprehensive coverage of the available evidence on multi-component exercise-based injury prevention programs in soccer.

While the inclusion of studies from the past decade provides a more current understanding of injury prevention interventions, the absence of a time restriction allows for the inclusion of earlier studies that may still offer valuable insights. Future research could benefit from narrowing the inclusion criteria to studies conducted within a specific time frame to account for advances in training methods and injury prevention strategies.

## 3. Results

### 3.1. Literature Identification

We initially identified 8944 records. After removing duplicates and excluding case reports and review articles, 2545 articles remained. Based on titles and abstracts screening, we evaluated 75 full-text articles for eligibility, with 15 meeting the inclusion criteria. These results are illustrated in Figure 1.

### 3.2. Demographic and Study Characteristics

All fifteen included studies were RCTs [17,19,24,25,26,41,42,43,44,45,46,47,48,49,50]. Of these, fourteen examined predominantly Caucasian populations, and one focused on an Asian population. Thirteen studies investigated adolescent populations, while two included participants with an average age over thirty. Five studies enrolled only female soccer teams, eight enrolled only male teams, and one included mixed-gender teams. These characteristics are summarized in Table 1.

### 3.3. Risk of Bias

A comprehensive risk-of-bias assessment was performed for all included RCTs. The results indicated a low risk of bias in the domains of random sequence generation and incomplete outcome data, suggesting high quality regarding both the randomization process and the management of missing data.

In contrast, moderate risks of bias emerged in allocation concealment, participant and personnel blinding, and outcome assessment blinding. These moderate risks could compromise the internal validity of the findings, as inadequate blinding may introduce both performance bias and detection bias.

Concerning other sources of bias, a potentially high risk was identified, possibly reflecting issues in study design, execution, analysis, or reporting that could systematically skew the results. Of the included studies, eight were classified as low risk of bias, one as high risk, and six showed some risk of bias in specific domains. These variations must be considered when interpreting the overall findings. A summary of these assessments is presented in Figure 2.

### 3.4. Lower Extremity Injuries

Eight RCTs evaluated the effectiveness of intervention programs in reducing lower extremity injuries among soccer players (Appendix A). As illustrated in Figure 3, the intervention group experienced 703 injuries among 5072 participants, whereas the control group recorded 943 injuries among 5011 participants. The pooled RR was 0.73 (95% CI: 0.63 to 0.84, *p* = 0.035), indicating a statistically significant reduction in lower extremity injuries for the intervention group. These findings suggest that the implemented intervention programs effectively decreased injury incidence compared with standard practice.

### 3.5. Hamstring Injuries

Four RCTs examined the impact of intervention programs on hamstring injuries in soccer players (Appendix A). As shown in Figure 4, the intervention group reported 51 hamstring injuries among 1722 participants, whereas the control group reported 62 injuries among 1518 participants. The pooled RR was 0.77 (95% CI: 0.37 to 1.61, *p* = 0.028), indicating a statistically significant reduction in hamstring injuries for the intervention group compared with the control group.

However, it is important to note that the 95% CI crosses one (0.37 to 1.61), which means that the true effect could be as large as a 61% increase in risk or as small as a 63% reduction in risk. Although the *p*-value suggests statistical significance (*p* = 0.028), the wide range of the CI implies some uncertainty about the precise effect size. This may indicate that while there is a trend toward a reduction in hamstring injuries, further studies with larger sample sizes or more homogeneous populations are needed to confirm the robustness and consistency of this effect. These results highlight the importance of interpreting statistical significance with caution, particularly when CIs are wide and include the possibility of no effect or even a harmful effect.

### 3.6. Knee Injuries

Thirteen RCTs evaluated the effectiveness of intervention programs on knee injuries among soccer players (Appendix A). As shown in Figure 5, the intervention group experienced 313 knee injuries out of 10,642 participants, whereas the control group experienced 488 knee injuries out of 10,297 participants. The pooled RR was 0.67 (95% CI: 0.54 to 0.84, *p* = 0.014).

### 3.7. Ankle Injuries

Eight RCTs assessed sports intervention programs on ankle injuries in soccer players (Appendix A). As shown in Figure 6, the intervention group experienced 252 ankle injuries among 6307 participants, while the control group reported 323 ankle injuries among 5856 participants. The pooled RR was 0.70 (95% CI: 0.57 to 0.86, *p* = 0.0173).

### 3.8. Publication Bias Assessment

A funnel plot is a graphical tool for detecting publication bias, where asymmetry or missing data can indicate potential bias. Figure 7 presents the funnel plots for evaluated outcomes, labeled A, B, C, and D. Except for one study, all studies on lower extremity injuries appear within the corresponding funnel plot, and separate funnel plots are provided for hamstring, knee, and ankle injuries. The overall symmetry of these plots suggests a minimal risk of publication bias.

Further assessment using Egger’s test yielded *p*-values of 0.395 for lower extremity injuries, 0.359 for hamstring injuries, 0.629 for knee injuries, and 0.674 for ankle injuries.

## 4. Discussion

This systematic review and meta-analysis provided a comprehensive synthesis of the existing evidence on the effectiveness of multi-component exercise-based injury prevention programs in soccer. The results underscore the significant impact of these programs in reducing the incidence of musculoskeletal injuries among soccer players, regardless of age or gender. The pooled data from the included RCTs reveal a substantial reduction in lower limb injuries, hamstring injuries, knee injuries, and ankle injuries, highlighting the efficacy of preventive measures in soccer injury management. These findings are consistent with those reported by other researchers [51]. The results of this study align with previous meta-analyses, which have shown a reduction in injury risk following the implementation of exercise-based prevention programs. The pooled RR of 0.73 for lower limb injuries is slightly lower than the injury incidence rate ratios reported in other studies, which may be attributable to the inclusion of a more diverse participant pool and a wider range of injury prevention programs [52,53]. Although multi-component exercise-based interventions are generally effective, our meta-analysis results suggest that the efficacy of these interventions may vary across different body regions and injury types, further highlighting the potential of multi-component intervention programs in preventing musculoskeletal injuries in soccer players.

Increased muscle strength helps improve overall body stability and resilience. In contact sports, strong muscles can effectively absorb external impacts, reducing soft tissue injuries caused by excessive stretching or twisting [54,55]. Improved joint stability enhances the strength of muscles around the joint, protecting against sprains in the knee or ankle. It also improves the athlete’s joint control, reducing the likelihood of injuries [56,57]. This suggests that certain components of the programs, such as strength and plyometric exercises, may also contribute to the reduction of contact injuries by improving functional stability and enhancing the ability to absorb external forces.

In female soccer athletes, a noticeable reduction in the occurrence of knee, ankle, and hip/groin injuries was observed, with reductions ranging from 15% to 17% for knee injuries, 17% to 22% for ankle injuries, and 25% to 29% for hip/groin injuries. However, further research must be conducted to enhance the accuracy and reliability of these findings. Notably, the observed reductions in injury rates for female athletes were lower than those documented in male athletes, where the reductions were more substantial, with knee injuries decreasing by 32% to 58%, ankle injuries by 30% to 60%, and hip/groin injuries by 47%. The discrepancies in injury reduction effectiveness between sexes warrant further investigation to identify underlying factors and refine injury prevention strategies accordingly [19,32,47].

While the current study focused on isokinetic quadriceps strength assessment, it is noteworthy that recent evidence challenges the clinical validity of isolated strength metrics in injury prediction [58]. Particularly in soccer athletes, the multifactorial nature of lower limb injuries necessitates an integrated evaluation encompassing neuromuscular control, movement asymmetry, and sport-specific fatigue patterns [59]. This methodological consideration will be prioritized in our forthcoming prospective cohort study design.

The analysis revealed variations in the effectiveness of prevention programs across different sexes and age groups. The reduced injury risk observed in children and youth, as compared to senior and veteran players, may reflect the relative novelty of preventive measures in younger populations and the potential for greater adherence and benefit. These findings highlight the need for tailored prevention strategies that take into account the specific physiological and biomechanical characteristics of different age groups [60].

## 5. Strengths and Limitations

This review and meta-analysis makes several key contributions to soccer injury prevention. It provides a comprehensive summary of RCTs across different genders and age groups, filling a gap in the current literature. The study’s rigorous methodology, following PRISMA guidelines, strengthens the reliability of its conclusions. A major finding is the strong evidence that multi-component exercise-based injury prevention programs reduce musculoskeletal injuries, particularly in the lower limbs, hamstrings, knees, and ankles. These results highlight the importance of such programs in professional soccer, potentially shifting injury management strategies. The study’s broad applicability to various demographic groups offers valuable insights for coaches, athletic trainers, and policymakers. However, there are some limitations: the variability across studies may affect the results, and the exclusion of non-RCT designs limits the scope. Also, the generalizability may be limited due to a focus on specific populations. The long-term adherence and effectiveness of the programs were not fully assessed, and while publication bias was found to be non-significant, it cannot be completely ruled out. Future studies should address these limitations by incorporating more targeted multicenter RCTs, evaluating long-term effectiveness, and broadening the coverage of diverse populations.

Publication bias is a critical factor to consider in systematic reviews and meta-analyses. In this study, we assessed publication bias and found no significant evidence of bias. Although these statistical tests suggest that publication bias is unlikely to significantly affect our findings, we recognize that these methods are not foolproof, and bias cannot be entirely excluded. Factors such as selective reporting, language bias, or the tendency for positive results to be published more frequently could still contribute to subtle biases in the literature. Additionally, the ability of statistical tests to detect publication bias may be limited by small sample sizes or incomplete reporting in the studies included. Thus, while our analysis suggests minimal bias, caution is warranted, and potential unaccounted biases should be considered when interpreting the results. Future studies could benefit from more comprehensive sensitivity analyses to further assess the impact of publication bias [61,62,63,64].

## 6. Conclusions

In conclusion, this systematic review and meta-analysis provides compelling evidence supporting the incorporation of multi-component exercise-based injury prevention programs into routine soccer training protocols. These interventions effectively reduce the risk of musculoskeletal injuries, thereby promoting musculoskeletal health and injury prevention among soccer players. The flexibility of these programs allows for customization according to the specific needs of individual players and teams, promoting both the overall health and performance of soccer athletes. The integration of such preventive measures into standard training practices holds significant potential for improving long-term player safety and contributing to the sustainability of high performance in the sport.

## Figures and Tables

**Figure 1 healthcare-13-00765-f001:**
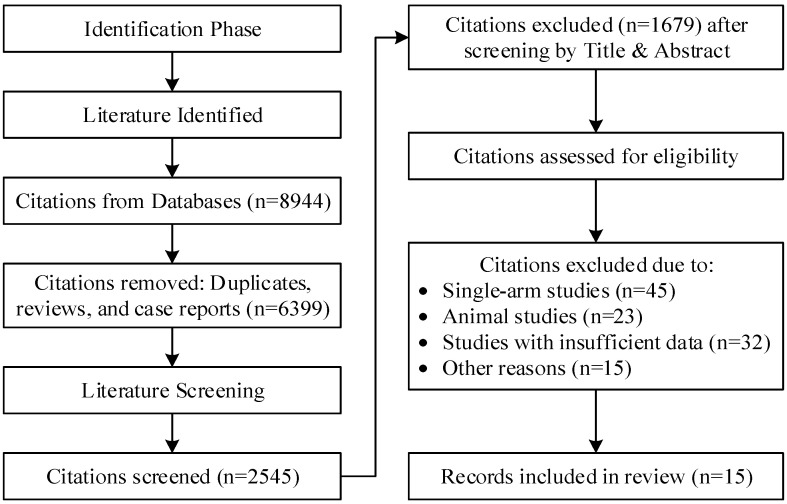
Flow chart of the included studies.

**Figure 2 healthcare-13-00765-f002:**
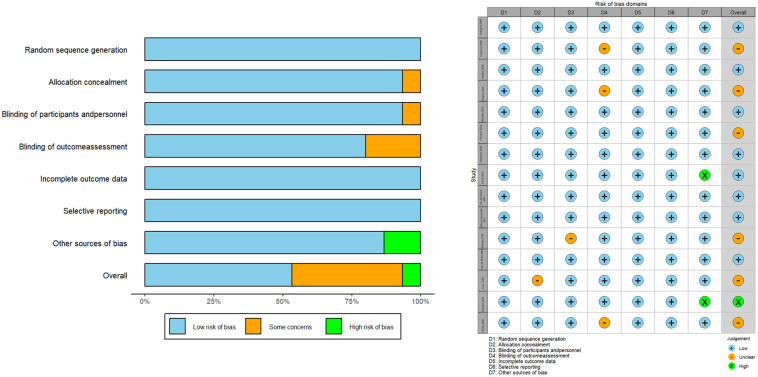
Overall bias chart and bias by studies [17,19,24,25,26,41,42,43,44,45,46,47,48,49,50].

**Figure 3 healthcare-13-00765-f003:**
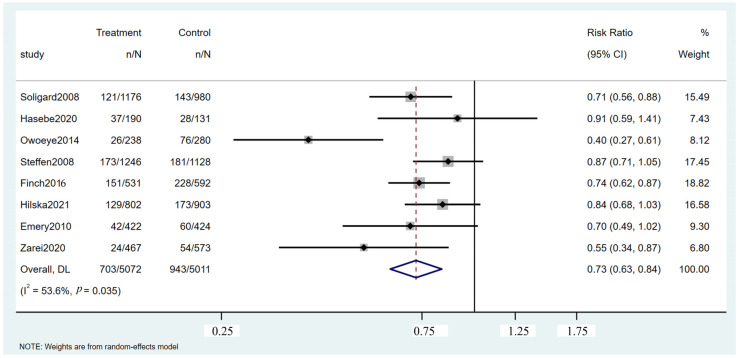
Forest plot of the lower extremity injuries [17,26,41,42,45,46,47,48].

**Figure 4 healthcare-13-00765-f004:**
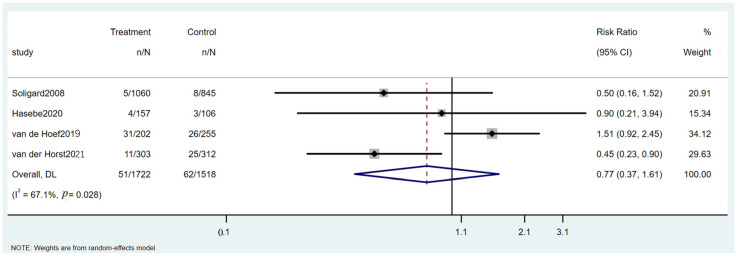
Forest plot of the hamstring injuries [17,24,45,49].

**Figure 5 healthcare-13-00765-f005:**
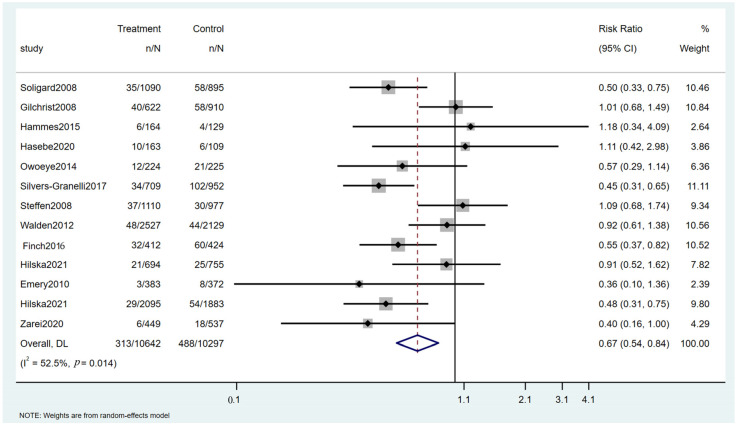
Forest plot of the knee injuries [17,19,26,41,42,43,44,45,46,47,48,50].

**Figure 6 healthcare-13-00765-f006:**
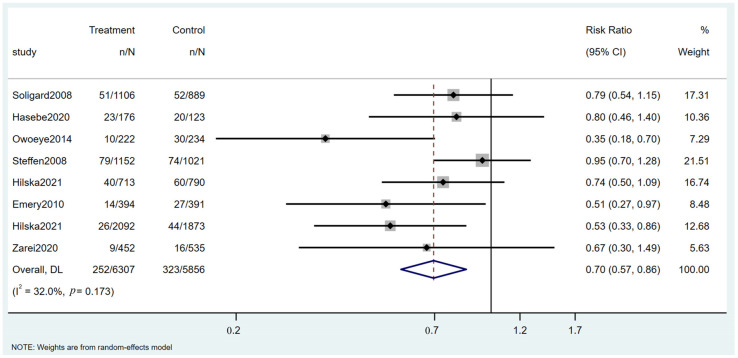
Forest plot of the ankle injuries [17,26,41,45,46,47,48].

**Figure 7 healthcare-13-00765-f007:**
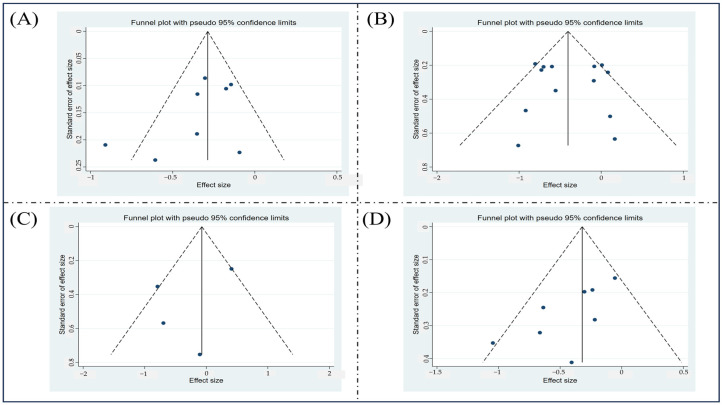
Funnel plots of the outcomes. (**A**) Lower extremity injuries; (**B**) hamstring injuries; (**C**) knee injuries; (**D**) ankle injuries.

**Table 1 healthcare-13-00765-t001:** Basic characteristics of the studies.

Study	Publication Years	Age (Years)	Males (%)	Females (%)	Intervention (n)	Control (n)	Outcomes
Soligard et al. [17]	2008	15.4 ± 0.7	0	100	General FIFA 11+ 2 times/week (1055)	Routine training (837)	Lower extremity injuries
Gilchrist et al. [43]	2008	19.9	0	100	General prevent injury and enhance performance (PEP) program 3 times/week (583)	Routine training (852)	Knee injuries
Steffen et al. [48]	2008	15.4 ± 0.8	0	100	General FIFA 11 (1073)	Routine training (947)	Lower extremity injuries, knee injuries, ankle injuries
Emery and Meeuwisse [41]	2010	17.2 ± 1.1	0	100	Neuro-muscular training program (380)	Routine training (364)	Lower extremity injuries, knee injuries, ankle injuries
Waldén et al. [50]	2012	14 ± 0.2	0	100	General neuro-muscular training (2479)	Routine training (2085)	Knee injuries
Owoeye et al. [47]	2014	17.6 ± 1	100	0	General FIFA 11+ 2 times/week (212)	Routine training (204)	Lower extremity injuries, knee injuries, ankle injuries
Hammes et al. [44]	2015	44.4 ± 7.7	100	0	FIFA 11+ (158)	Routine training (125)	Knee injuries
Finch et al. [42]	2016	33.4 ± 4.5	100	0	Neuro-muscular training program (380)	Routine training (364)	Lower extremity injuries, knee injuries
van der Horst et al. [24]	2021	24.5 ± 3.8	100	0	Nordic hamstring exercise (292)	Routine training (287)	Hamstring injuries
Silvers-Granelli et al. [19]	2017	21.5 ± 1.5	100	0	General FIFA 11+ 2–3 times/week (675)	Routine training (850)	Knee injuries
Rössler et al. [25]	2018	10.7 ± 1.4	NA	NA	FIFA 11+ (2066)	Routine training (1829)	Knee injuries, ankle injuries
van de Hoef et al. [49]	2019	23.4 ± 4.5	100	0	Bounding exercise program (171)	Routine training (229)	Hamstring injuries
Zarei et al. [26]	2020	12.2 ± 1.8	100	0	FIFA 11+ (443)	Routine training (519)	Lower extremity injuries, knee injuries, ankle injuries
Hasebe et al. [45]	2020	16.5 ± 0.6	100	0	The Nordic hamstring exercise (153)	Routine training (103)	Lower extremity injuries
Hilska et al. [46]	2021	12.3 ± 1.2	81	19	Neuro-muscular training program (673)	Routine training (730)	Lower extremity injuries, knee injuries, ankle injuries

## Data Availability

The data that support the findings of this study are available on reasonable request from the corresponding author. The data are not publicly available due to privacy or ethical restrictions.

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
