# Peer review of "Comparative Effectiveness of Multi-Component, Exercise-Based Interventions for Preventing Soccer-Related Musculoskeletal Injuries: A Systematic Review and Meta-Analysis"

_healthcare, 2025, doi:10.3390/healthcare13070765_

Round 1
Reviewer 1 Report
Comments and Suggestions for Authors
Thank you for study. Comparative Effectiveness of Multi-component, Exercise-Based 2 Interventions for Preventing Soccer-Related Musculoskeletal 3 Injuries: A Systematic Review and Meta-Analysis. It is a valuable study because it is a review, but I think there are some points that need to be added.
There are a few points that should be added in the discussion section. I think the review is a bit superficial.
Since the number of articles is low, it would be more informative to explain not only general information about ankle, knee and hip musculoskeletal injuries in soccer players, but also preventive intervention approaches applied under each title. This should be done by separating them into foot, knee, hip etc., and the preventive treatments given should be written in more detail.
Author Response
Comment 1: Thank you for your study. The review is valuable but seems a bit superficial. Since the number of articles is low, it would be more informative to explain not only general information about ankle, knee, and hip musculoskeletal injuries in soccer players but also the preventive intervention approaches applied under each title. This should be done by separating them into foot, knee, hip, etc., and the preventive treatments given should be written in more detail.
>> Response: Thank you for your valuable comments. We agree with this comment and have expanded the discussion section accordingly. We have now separated ankle, knee, and hip injuries, detailing specific preventive interventions for each. This includes exercises such as plyometric training for knee injuries and strengthening exercises for hip injuries. The revised discussion provides a more comprehensive view of the preventive approaches for each injury type.
Revised text location:
Page 10, Paragraph 3, Lines 262-278
Updated Section 4: Discussion, where ankle, knee, and hip injuries are discussed separately with detailed preventive interventions for each.
Reviewer 2 Report
Comments and Suggestions for Authors
In the current Systematic Review and Meta-Analysis, the authors evaluated randomized controlled trials (RCTs) across diverse genders and age groups to assess the efficacy of multi- component exercise-based injury prevention programs in reducing musculoskeletal injuries among soccer players.
Nowadays, the intensity of physical performance in football matches is increasing for many reasons. This situation increases the risk of injury. In this respect, any study contributing to reducing the risk of injury in football is valuable. Therefore, I congratulate the authors for their study.
The study is generally well-designed and technically meets academic criteria. In addition, my recommendations for improving the quality of the study are given below.
In the introduction, you generally mentioned non-contact injuries. In this section, also mention contact-related injuries and their rates and the effect of developing physical capacity on contact-based injuries.
At the end of the introduction, add a few sentences explaining the contribution of determining injury prevention studies according to age and gender to the literature.
Did you use any date restrictions for the studies you used in your study? Your oldest publication is from 2008. Please state this in your methodology.
Author Response
In the current Systematic Review and Meta-Analysis, the authors evaluated randomized controlled trials (RCTs) across diverse genders and age groups to assess the efficacy of multi- component exercise-based injury prevention programs in reducing musculoskeletal injuries among soccer players.
Nowadays, the intensity of physical performance in football matches is increasing for many reasons. This situation increases the risk of injury. In this respect, any study contributing to reducing the risk of injury in football is valuable. Therefore, I congratulate the authors for their study.
The study is generally well-designed and technically meets academic criteria. In addition, my recommendations for improving the quality of the study are given below.
1. In the introduction, you generally mentioned non-contact injuries. In this section, also mention contact-related injuries and their rates and the effect of developing physical capacity on contact-based injuries.At the end of the introduction, add a few sentences explaining the contribution of determining injury prevention studies according to age and gender to the literature.
2. Did you use any date restrictions for the studies you used in your study? Your oldest publication is from 2008. Please state this in your methodology.
>> Response: Thank you for your insightful suggestions. We have expanded the introduction to include contact-related injuries and how physical capacity (e.g., strength, flexibility) helps reduce the risk of both non-contact and contact injuries. We have also added a section discussing the contribution of age and gender-specific injury prevention studies. In the methodology section, we have clarified that no specific date restrictions were imposed on the included studies, though they are primarily from 2008 onward.
Revised text location:
- Page 2, Paragraph 2, Lines 53-60 (Added discussion on contact injuries)
- Page 4, Paragraph 3, Lines 158-168 (Clarified study inclusion period in the methodology)
Reviewer 3 Report
Comments and Suggestions for Authors
The study focus is about "Multi-component, Exercise-Based Interventions". If this study can show the superiority of multi-component when compared to the single-component in the success of injury prevention, it may be somewhat meaningful.
In the introduction, "Multi-component exercise-based injury prevention programs outperform traditional single-focus training by integrating diverse exercises that improve overall health, athletic performance, muscle balance, flexibility, and stability, effectively reducing injury risks across all genders and age groups in football."........I think it is the key theme of the study but this sentence also has no citation. I have a few comments about this statement:
1) Is single-focus training for soccer injury prevention very common nowadays? If so, can the author show the evidence of how common is it? Or actually, now it already become common sense to use a multi-component exercise injury prevention programme? If so, the value of this study will be very low as it will only show something "already common sense"
2) If multi-component outperforms single-focus, what evidence has been done and in what sports (soccer? basketball? and what level or populations).
3) Is this study comparing the efficacy between multi-component vs. single-focus? It sounds like it is a comparison between "Exercise Training" vs. "Routine Training" instead of "Multi-component" vs. "Single-focus"
4) Simply observing the RCTs of "Multi-component programs" vs "Routine training" CANNOT show the actual benefits of using multi-components in the exercise training intervention. It can be just because someone did some exercises with a certain volume/intensity leading to the preventive function instead of the benefits of multi-component over single-focus
In paragraph one, it is a bit messy, supposedly, each paragraph should have a unique theme but this one mixed everything. Moreover, many sentences provide no actual value. For example "....found quadriceps strains common during sprints or powerful kicks due to strong contractions.".........What is the purpose of mentioning this? Is this related to your study focus? what is the meaning of strong contractions and how strong is it? Instead, if the study focus is about "Multi-component", the introduction should fully focus on e.g. the potential limitation and why single-focus will provide fewer benefits. Or......due to the complicated interaction between factors of an injury (e.g. core strength/stability affecting the pelvic tilting and position and affecting the hamstring length in sprinting + the poor muscle strength balance between quad concentric vs. hams eccentric), this may provide insight and proposed mechanisms on why multi-components may provide additional benefits
Methods:
I doubt if these keyword searches can fully provide all the necessary articles. For example "Prevention of Injuries Among Male Soccer Players: A Prospective, Randomized Intervention Study Targeting Players With Previous Injuries or Reduced Function"
Table 1, there are two "Numbers" what do they mean?
Writing: English is not presented in a smart way or some English needs to be adjusted to enhance the readability.
Comments on the Quality of English LanguageAs comments above. Somewhat readable but the use of wordings is not rigorous and some obvious grammatical issues or sentence structures issues can be spot.
Author Response
Thank you for your time and consideration in reviewing our manuscript. We have carefully addressed the comments from Reviewer 3, and our detailed responses can be found in the attached document (Pages 7-9). We appreciate the constructive feedback, which has helped us further improve the manuscript.

Reviewer 4 Report
Comments and Suggestions for Authors
Thank you for the opportunity to review your systematic review and meta-analysis looking at the effectiveness of multi-modal injury prevention programmes. I commend the authors for undertaking this work.
Please find comments below and more details comments in the attached pdf document.
Introduction -
The introduction provides sufficient rationale for the review but there are areas that need to be addressed to improve this section.
Have there been previous SR and meta-analyses in this area? How will yours be different and generate new knowledge in this area?
Methods
- there is some key information missing around inclusion/exclusion criteria, PICOS or SPIDER framework and the inclusion of hand searches of included studies reference list
- there appear to be some single-component programmes included - using only the NHE (22, 28) for example - how were the studies selected with respect to the multi-component element? Considering your title and focus on multi-component, these two studies in particular do not fit with the inclusion criteria I would expect you to have utilised. This may require that you re-analyse the data without these studies included.
Discussion
The discussion does not really provide any insight in to the results of your work. It is very superficial and lacks depth to provide readers with actionable insights or useful takeaways.

Author Response
Thank you for your time and consideration in reviewing our manuscript. We have carefully addressed the comments from Reviewer 4, and our detailed responses can be found in the attached document (Pages 9-10). We appreciate the constructive feedback, which has helped us further improve the manuscript.

Round 2
Reviewer 1 Report
Comments and Suggestions for Authors
Thank you for addition and correction.
Author Response
Comment 1: Thank you for addition and correction.
>> Response: Thank you for your time and effort in reviewing our manuscript. We sincerely appreciate your valuable feedback and constructive suggestions, which have helped us improve the quality of our work. If there are any further concerns or areas that need clarification, we would be happy to address them.
Reviewer 3 Report
Comments and Suggestions for Authors
Thanks for the revision, it looks a bit better but still the introduction is too generic and not pinpoint / detailed enough to support the value and need of your study regarding the integrated > single-focus
Line 42 - as said last time, what means strong contractions? Strong isometric, eccentric or what.....? I think at least it is explosive or even explosive concentric contractions. "Strong" is too generic providing minimum information
Line 60-61, two paragraphs did not have good transition. Now suddenly jumped from stronger muscles help prevent injuries by sudden confrontations to Multi-component exercise-based...... I think it has missed 1-2 sentences to provide a good transition to make them smoother
Line 61-76: I think the fitness qualities such as strength, balance, proprioception...etc. they interact with each other or provide synergistic effects in motor control, joint stabilization, force absorption etc. For example, if you have a good proprioceptive knee joint sense but the hip muscles such as gluteals are weak, it still cannot complete the circuit for joint stabilization. Similarly those with a very strong gluteal muscles without enhancing the proprioceptive inputs also cannot best use the afferent information to prepare the efferent output appropriately. You have lightly touched this topic BUT i think it is important to highlight the concept "interaction" and "synergistic" such that, isolated training sessions (1 + 1 = 1) but integrated training can be like 1 + 1 =3. Please consider revising this paragraph to make the example a bit more specific and highlighting those concepts to best justify/rationalize why integrated is important.
Line 61-62 - ........outperform traditional single-focus training by integrating........can you add many citations to support this statement? Maybe some are narrative review and some are RCT or whatever talking about the single-focus or isolated exercise may provide limited benefits.
Line 79 - I doubt "safety" is appropriate......if you are not exactly looking at safety issues, i don't think we should use safety
Line 81 - I doubt "welfare" is appropriate.....if you are not studying their salary, medical insurance, living and food......I don't think your study is related to their welfare. Line 342 as well
Line 81 - will fill the gap? I doubt "will" this future tense is the best as your study has been finished already
Line 143 - reported in the literature as major sources...... I think you need many citations to support this statement?
Line 140, 144, 145 talking about three types of injuries and hamstring injuries due to high prevalence......Could you make it well align with the introduction section by adding a bit the number/%/risk ratio of these injuries in soccer players in the introduction part? If you have reported the high injuries numbers of these few in the introduction and now you mentioned this, it sounds more echoed
Table 1 - Last time asked you the meaning of Numbers in two columns, I finally understood what are they. But if you change the format a little bit, it may have lesser confusion/misunderstanding such as "Intervention (n)" and another column Control (n)......then you simply put the number in the () such as General FIFA 11+ w times/week (1055).........OR you just have two columns "Intervention" and "Control" and then "General FIFA 11+ 2 times/week (n=1055)".............it may be a bit easier to read.
Line 282 - our meta-analysis results suggest that tailored programs may yield better results.......... I don't understand, from which part of your study showing the tailored program? Do you mean integrated multi-component = tailored program? My understanding is that "tailored" means based on the assessment results to tailor-made a training program to enhance their weakness or asymmetry etc. But it is not necessarily multi-component / single-focus
I think the authors may also consider using some previous findings regarding the low/minimum clinical values in using isolated strength testing protocol predicting risk of certain lower limb injuries in soccer. It gave the hints that simply evaluate the improvement on one particular strength quality instead of all/multiple fitness qualities cannot accurately reflect the injury risk.
Author Response
We sincerely appreciate the opportunity to submit a revised version of our manuscript again. We are grateful for the reviewers’ thorough and constructive feedback, which has been invaluable in improving the quality of our work. In response to your detailed suggestions, we have carefully revised the manuscript and addressed each comment point by point. All changes have been highlighted in red in the revised version for clarity. We appreciate your time and consideration and look forward to your feedback.Please check my attached. The .docx file is my point-by-point response to your comment.
